# Citrus Bright Spot Virus: A New Dichorhavirus, Transmitted by *Brevipalpus azores*, Causing Citrus Leprosis Disease in Brazil

**DOI:** 10.3390/plants12061371

**Published:** 2023-03-20

**Authors:** Camila Chabi-Jesus, Pedro Luis Ramos-González, Aline Daniele Tassi, Laura Rossetto Pereira, Marinês Bastianel, Douglas Lau, Maria Cristina Canale, Ricardo Harakava, Valdenice Moreira Novelli, Elliot Watanabe Kitajima, Juliana Freitas-Astúa

**Affiliations:** 1Escola Superior de Agricultura Luiz de Queiroz, Universidade de São Paulo (Esalq/USP), Piracicaba 13418-900, São Paulo, Brazil; 2Instituto Biológico/IB, São Paulo 04014-002, São Paulo, Brazil; 3Centro de Citricultura Sylvio Moreira/IAC, Cordeirópolis 13490-970, São Paulo, Brazil; 4Embrapa Trigo, Passo Fundo 99050-970, Rio Grande do Sul, Brazil; 5Empresa de Pesquisa Agropecuária e Extensão Rural de Santa Catarina/Epagri, Paulo Lopes 88490-000, Santa Catarina, Brazil; 6Embrapa Mandioca e Fruticultura, Cruz das Almas 44380-000, Bahia, Brazil

**Keywords:** *Mononegavirales*, *Rhabdoviridae*, vector mites

## Abstract

Citrus leprosis (CL) is the main viral disease affecting the Brazilian citriculture. Sweet orange (*Citrus sinensis* L. Osbeck) trees affected by CL were identified in small orchards in Southern Brazil. Rod-like particles of 40 × 100 nm and electron lucent viroplasm were observed in the nucleus of infected cells in symptomatic tissues. RNA extracts from three plants, which proved negative by RT-PCR for known CL-causing viruses, were analyzed by high throughput sequencing and Sanger sequencing after RT-PCR. The genomes of bi-segmented ss(−)RNA viruses, with ORFs in a typical organization of members of the genus *Dichorhavirus*, were recovered. These genomes shared 98–99% nt sequence identity among them but <73% with those of known dichorhavirids, a value below the threshold for new species demarcation within that genus. Phylogenetically, the three haplotypes of the new virus called citrus bright spot virus (CiBSV) are clustered with citrus leprosis virus N, which is a dichorhavirus transmitted by *Brevipalpus phoenicis* sensu stricto. In CiBSV-infected citrus plants, *B. papayensis* and *B. azores* were found, but the virus could only be transmitted to Arabidopsis plants by *B. azores.* The study provides the first evidence of the role of *B. azores* as a viral vector and supports the assignment of CiBSV to the tentative new species *Dichorhavirus australis*.

## 1. Introduction

*Dichorhavirus* is a genus of the family *Rhabdoviridae* (order *Mononegavirales*) that groups plant-infecting viruses transmitted by mites of the genus *Brevipalpus* [1,2]. Dichorhaviruses have a bi-segmented negative-sense, single-stranded RNA genome (-ssRNA) [3]. RNA1, with ~7 kb, has five ORFs encoding the nucleocapsid protein (N), phosphoprotein (P), movement protein (MP), matrix protein (M), and glycoprotein (G). RNA2, ~6 kb, comprises a single ORF that codes for the RNA-dependent RNA polymerase (RdRp), also known as the L protein.

Viruses of the five species of dichorhavirus, recognized by the International Committee on Taxonomy of Viruses (ICTV) (https://talk.ictvonline.org/taxonomy/) (accessed on 10 December 2022) have been detected infecting crops such as citrus (*Citrus* spp.), coffee (*Coffea* spp.), and several species of ornamentals such as glory-bower and orchids [4]. Additionally, dichorhavirus-like particles that have been observed by transmission electron microscopy (TEM) in ornamental plants, e.g., *Cestrum nocturnum* L., *Solanum violifolium* Schott, and *Vinca major* L., are not detected by RT-PCR tests using specific primers for known dichorhaviruses, suggesting a larger and undiscovered diversity of viruses potentially belonging to this genus [5,6].

Dichorhaviruses are plant-to-plant transmitted by some species of flat mites of the genus *Brevipalpus* (Acari: Tenuipalpidae), where they can also replicate [7]. Orchid fleck virus (OFV, *Dichorhavirus orchidaceae*) is transmitted by *B. californicus* to orchids and citrus [6,8], and it causes citrus leprosis (CL) disease in Mexico, Colombia, South Africa, and Hawaii (USA) [9,10,11,12]. In Brazil, OFV has been found infecting orchids but not citrus, which is likely due to the absence of *B. californicus* in citrus orchards [13]. However, the dichorhaviruses citrus leprosis virus N (CiLV-N, *Dichorhavirus leprosis*), transmitted by *B. phoenicis* sensu stricto (s. s.), and citrus chlorotic spot virus (CiCSV, *Dichorhavirus citri*), associated with *B. yothersi* and *B*. aff. *yothersi,* cause CL in Brazil [14,15,16]. The relationship between *Brevipalpus* spp. and the dichorhaviruses they transmit is often very strict, likely as a consequence of coevolution between them, as is suggested by phylogenetic data [14].

CL is considered the main viral disease that affects citrus production in Brazil [17,18]. The disease can be caused by viruses of at least five species within the genera *Dichorhavirus* and *Cilevirus* [19], but commercial orchards in Brazil are almost exclusively affected by citrus leprosis virus C (*Cilevirus leprosis*, genus *Cilevirus*, family *Kitaviridae*) [20]. Differently from Mexico and Colombia [8], CL caused by dichorhaviruses in Brazil has been reported only in marginal citrus-growing areas [16,21]. CL causes localized chlorotic and/or necrotic lesions in leaves and fruits, and it can induce dieback, premature leaf drop, and often leads to plant death [19]. The control of *Brevipalpus* mites in orchards of the Southeastern region of Brazil, the main citrus-producing area of the country, represents almost 5% of the management cost of the orchards, i.e., approximately US $54 million per year [17]. In that region, premature fruit drop associated with CL was assessed in 5.82 million boxes in the 2020/2021 harvest (https://www.fundecitrus.com.br) (accessed on 20 July 2022) [17,18,22]. The northeast and southern regions represent 10% and 8% of citrus national production, respectively (https://www.ibge.gov.br/) (accessed on 20 July 2022), but the epidemiology of CL and losses associated with this disease remain poorly known [23].

In this study, we analyzed sweet orange (*Citrus sinensis* L. Osbeck) tree samples showing typical symptoms of CL collected in three small citrus orchards from the southern region of Brazil. We recovered the genomic sequence of three new viral isolates, which likely belong to a new species of plant-infecting rhabdovirus of the genus *Dichorhavirus*. Additionally, we show evidence of viral transmission by a species of *Brevipalpus* mites previously unreported as a virus vector. A specific method based on RT-PCR for the detection of the new dichorhavirus is also described.

## 2. Results

### 2.1. Identification of Rod-like Particles Associated with Citrus Leprosis Symptoms

Although brighter, lesions observed in the leaves of the sweet orange plants collected in the states of *Santa Catarina* (SC) and *Rio Grande do Sul* (RS), Brazil resembled chlorotic spots described in citrus plants affected by CL disease (Figure 1). Necrotic lesions were frequently found in the fruits but not in the affected leaves (Figure 1). TEM analyses of ultrathin sections of the lesions on symptomatic leaves collected in Passo Fundo and Marquês de Souza, RS, and Seara, SC, revealed cytopathic effects of infection by dichorhaviruses, characterized by the electron-lucent viroplasm in the nucleus and the presence of rod-like particles of ~40 × 100 nm, dispersed or aggregated in the nucleoplasm and forming the so-called “spoke wheel” arrangement in the cytoplasm of the parenchymal cells in the foliar lesions (Figure 2) [3]. Molecular tests for the detection of all known dichorhaviruses and cileviruses produced negative results (Figure 2 and data not shown).

### 2.2. Characterization of Three Genome Sequences Confirms the Infection by Putative Novel Plant Rhabdoviruses

A total of 12,633,399 and 12,884,461 reads were recovered from the high throughput sequencing (HTS) libraries prepared from the sweet orange RNA extracts of the samples MSo01 and Ser01, respectively. BlastX searches of the assembled contigs, using a customized local database of non-redundant viral proteins, resulted in contigs with 3893 and 2743 nts from the sample MSo01, as well as 1863 and 4813 nts from the sample Ser01. These contigs contained sequences corresponding to ORFs that encode proteins sharing 71–85% amino acid sequence identity with proteins N, P, MP, M, and G of the dichorhavirus CiLV-N [RefSeq accession number NC052230). Besides, one contig with 6012 nts from the MSo01 library and three contigs with 227, 2449 and 3273 nts, from the sample Ser01, were recognized due to their best identity values (~74%) with the L protein of CiLV-N (RefSeq NC052231). Sanger sequencing of partially overlapping amplicons, generated using specific primers, yielded sequences 99% identical to those from the HTS contigs, and allowed filling the remained gaps between the HTS-generated contigs. Moreover, the PFd01 sample was analyzed by RT-PCR using specific primers for the isolates MSo01 and Ser01, and the assemblage of the generated sequences revealed the two genomic segments of a third viral isolate. The 5′ and 3′ ends of the two molecules of each isolate were determined by RACE.

Altogether, the genomes of the three isolates comprise ~13 kb divided into ~7 kb in RNA1 and ~6 kb in RNA2. Nucleotide sequence identity values, in paired comparisons between the genomic segments of the three isolates, were higher than 97%, whereas the values ranged from 98 to 100% in the analyses using the deduced amino acid (aa) sequences (Figure 3, Appendix A). The three isolates were considered variants of the same virus, hereafter called citrus bright spot virus (CiBSV). According to the number of plants infected and the availability of samples for future analyses, the isolate PFd01 was considered as the reference for further comparative analyses in this study.

Nucleotide sequence comparisons of CiBSV_PFd01 with other dichorhaviruses showed identity values in the range of 47–74% (Appendix A), and the highest values corresponded to the alignments with CiLV-N (Figure 3). Amino acid sequence identity values of the deduced proteins from the isolate PFd01 with those of other dichorhaviruses, ranged from 73 to 83% with proteins of CiLV-N, 33–63% with OFV (NC009609 and NC009608), 40–64% with CoRSV (NC038756 and NC038755), 38–64% with ClCSV (NC043648 and NC043649), and 39–64% with CiCSV (NC055208 and NC055209) (Appendix A). The genomic sequences of the identified viruses were deposited in the GenBank (GB) database under accession numbers CiBSV_PFd01: MZ773933 and MZ773938, CiBSV_MSo01: MZ773934 and MZ773936, and CiBSV_Ser01: MZ773935 and MZ773937.

No recombination events were found across the sequences of CiBSV_PFd01, CiBSV_MSo01, and CiBSV_Ser01; thus, they were used in phylogenetic analyses. Phylogenetic reconstruction, using N and L proteins of definitive and tentative members of the genus *Dichorhavirus* (Figure 4 and Appendix A), revealed that the isolates of CiBSV clustered with CiLV-N, which is a virus transmitted by *B. phoenicis* sensu stricto [16,24,25]. Reassortment events among dichorhavirus genomic segments of the subgroup 2, comprising CiLV-N and CiBSV, were not detected (Appendix A).

According to in silico genome annotation, N, P, G, and L proteins of CiBSV isolates showed conserved domains described for rhabdoviruses (https://www.genome.jp/tools/motif/) (accessed on 5 January 2022) (Appendix A). Besides, in silico analyses of the putative glycoprotein (G) of CiBSV indicated the presence of a signal peptide in the N-terminal and a transmembrane domain in the C-terminal (Appendix A), as regularly observed in other rhabdoviruses [26,27]. In addition, a homolog UL11 domain (PF11094, membrane-associated tegument protein) was found in the G protein of isolate PFd01. An in-depth description of viral predicted proteins is shown in Appendix A.

### 2.3. Brevipalpus Azores Mites Are Able to Transmit CiBSV Isolates

There were six *Brevipalpus* mite specimens collected in leaves and branches of the sweet orange trees of the samples MSo01, MSo02 and Ser01, and they were used directly for morphological identification. In the PFd01 and PFd02samples, more than 50 specimens of the genus *Brevipalpus* were found, so these mites were used for morphological identification, molecular characterization, and transmission assays. The partial nucleotide sequences of the COI gene, from two *Brevipalpus* specimens collected in the sample PFd02 (GB accession number OQ380524) and MSo01, shared >99.7% nucleotide identity with reference sequences of *Brevipalpus azores* (GB accession number MK499460). Besides, morphoanatomical evaluation of another set of mites collected from all symptomatic samples revealed: (i) on the prodorsum central cuticle foveolated, a sublateral cuticle with some rounded cells posteriorly; (ii) opisthosoma with six pairs of dorsolateral setae, setae *f2* absent, and cuticle between *c1*-*c1* to *e1*-*e1* smooth or with some weak wrinkles and cuticle between *e1*-*e1* to *h1*-*h1* with a series of transverse folds; (iii) ventral shield cuticle with weak transverse bands; (iv) genital plate cuticle with uniform narrow transverse bands; (*v*) palps four was segmented; (vi) tarsus II with two short rod-shaped solenidia, which matched the features characterizing members of the species *B. azores,* previously misidentified as *B. phoenicis* sensu lato (s.l.) [25] (Figure 5). In the samples PFd01 and PFd02, 2 out of 34 collected mites used for identification were of the species *B. papayensis* [25], but those were not encountered amongst the specimens used on the transmission assays.

*B. azores* mites, collected in the CiBSV-infected citrus sample from PFd02, were transferred to 10 healthy *Arabidopsis thaliana* plants. Leaf samples from two of the infested plants showed chlorotic symptoms and tested positive in RT-PCR using specific primers for the detection of the *N* and *L* genes of CiBSV (Appendix A). Partial sequences of *N* (GB accession numbers OQ601569 and OQ601571) and *L* (OQ601570 and OQ601572) genes, obtained from the two infected Arabidopsis plants, shared more than 99% nucleotide sequence identity with the sequences of CiBSV_PFd01 isolate. Attempts to transmit CiBSV to sweet orange trees, Arabidopsis, and common bean (*Phaseolus vulgaris* L.) plants, originally using non-viruliferous *B. papayensis, B. obovatus, B. californicus* s.l., and *B. yothersi* mites that fed onto leaves of the sample PFd02, were unsuccessful. Plants remained asymptomatic and tested negative in RT-PCR assays for the detection of CiBSV (data not shown).

### 2.4. Primers for the Detection of CiBSV Are Specific

In addition to the detection of CiBSV-infected plants, the performance and specificity of the pair of primers, *N*F-*N*R, were also tested on RNA extracts obtained from plants infected with the dichorhaviruses CiLV-N, CiCSV, ClCSV, CoRSV, and OFV. As expected, amplicons of approximately 296 bp were observed only from sweet orange samples infected with CiBSV (Figure 6), which were collected in MSo02, Ser01, and PFd02. These amplicons (GB accession numbers OQ601573 and OQ601574) shared more than 98% nucleotide sequence identity with the sequences of the isolate CiBSV_PFd01.

## 3. Discussion

The prevalent causal agent of citrus leprosis (CL) disease in commercial orchards in Latin America is citrus leprosis virus C (*Cilevirus leprosis*, family *Kitaviridae*) [20], yet CL-producing dichorhaviruses, also transmitted by *Brevipalpus* spp., have been identified in commercial and non-commercial citrus areas of the Americas and South Africa, but they are not normally of economic concern [11,12,16,24]. However, eventually, under the influence of emerging climatic changes and in the absence of strict control measures, these dichorhaviruses could jeopardize orchard health and reduce citrus yield.

In this study, we identified three isolates of a novel dichorhavirus causing CL in citrus trees in southern Brazil (Figure 1). Large chlorotic spots of bright yellow color, mostly rounded and similar to the symptoms induced by the dichorhavirus CiCSV [14], were observed in sweet orange leaves. The lesions were different, however, from those caused by CiLV-N, often characterized by the appearance of necrotic spots with chlorotic halos [16], or by OFV, which causes both chlorotic and/or necrotic lesions [8,11,12]. Suspicion of the involvement of a dichorhavirus, raised by cytopathological studies, was confirmed by subsequent molecular analyses, which indicated a new tentative dichorhavirus as the causal agent. Although symptoms in plants infected by different dichorhaviruses appear to show somewhat particular patterns, the use of these characteristics to identify the causal agent is imprecise. Therefore, the development of specific detection methods is essential for the proper identification and management of CL-causing viruses, aiming at reducing and preventing the spread of CL disease.

Based on the similar genomic organization among the studied isolates, nucleotide and amino acid sequence identity values higher than 98%, and their close phylogenetic relationships, we concluded that they belong to the same virus, representing a new species of dichorhavirus. Identities of CiBSV isolates with other dichorhaviruses are below the threshold for the designation of new species in the genus, i.e., less than 80% of nt sequence identity, in comparison, involving the complete sequence of the RNA1 and ORF *L* in the RNA2 [3]. Accordingly, we propose *Dichorhavirus australis* as the binomial name for the newly detected species [28]. The name of the tentative new species has been suggested while considering the current distribution of the virus in the southern region of Brazil.

CiBSV was transmitted to *A. thaliana* plants using viruliferous *Brevipalpus azores* mites. The assay demonstrated the involvement of a novel *Brevipalpus* species acting as a viral vector. However, considering that only two Arabidopsis plants were infected by CiBSV, the test may also suggest that (i) these mites inefficiently transmit CiBSV, and/or (ii) CiBSV reaches low titer in these plants. Transmission assays also proved that *B. papayensis*, *B. obovatus,* and *B. yothersi* are, at least under our experimental conditions, unable to transmit CiBSV isolates. It should be noted that these experiments were carried out in conditions equivalent to those commonly described for the transmission of other *Brevipalpus*-transmitted viruses [7,29]. Further studies could be performed using citrus seedlings as test plants to assess the transmission efficiency of CiBSV by *B. azores*.

*B. azores* mites were first identified in the Azores islands, Portugal [25]. Previous to this study, specimens of this species had been reported in twelve regions of the world but not in South America on plants of the families Araliaceae, Musaceae, Rubiaceae, Rutaceae, Solanaceae, Theaceae, and Vitaceae. In Rutaceae, specimens of *B. azores* were found in samples of citrus, e.g., tangerine (*Citrus reticulata* L.), from Portugal, which were intercepted in the USA in the period from 1975–1982 [25]. Mites of this species belong to the *B. phoenicis* sensu lato group [25]. Besides *B*. *azores*, the group is composed of seven other species, including *B*. *papayensis*, *B*. *yothersi,* and *B. phoenicis* s.s. Preliminary results on the phylogeny of *Brevipalpus* have demonstrated the existence of phylogenetic subgroups in which, interestingly, *B. phoenicis* s.s. and *B*. *azores* are clustered in the same clade (A. D. Tassi, personal communication).

Phylogenetic trees of dichorhaviruses have revealed three lineages [14,24] called subgroups 1, 2, and 3 (Figure 4). It is speculated that this topology likely reflects a virus–vector coevolutionary relationship [3]. Subgroup 1 is composed of CoRSV, ClCSV, and CiCSV, which, together, are vectored by more than one species of *Brevipalpus*. In common, these three viruses can be transmitted by *B. yothersi* mites, whereas CoRSV is also transmitted by *B. papayensis*, and CiCSV is vectored by *B.* aff. *yothersi* [14,24,30]. OFV stands alone in subgroup 2 and is the only dichorhavirus reported, so far, to be vectored by *B. californicus* [6]. Subgroup 3 is comprised of CiLV-N, transmitted by *B. phoenicis* s.s. [16], and phylogenetic analyses described in this study indicated that CiBSV shares the same clade with CiLV-N. Since infection by CiBSV is associated with the presence of *B. azores*, which also transmitted the virus to Arabidopsis, and *Brevipalpus*, phylogenetic studies cluster *B. phoenicis* s.s. and *B. azores* in the same clade, the result presented in this study may provide further support to the hypothesis of dichorhavirus-mite coevolution.

Specific interactions between dichorhavirus and mites can also account for the distribution of these viruses (Figure 4). Except for *B. yothersi*, *Brevipalpus* mites that vector CL-causing dichorhaviruses are rarely found in commercial citrus orchards in Brazil [14,16,31,32]. *B. californicus* was only reported once on citrus plants grown more than 3000 kilometers away from the main citrus-producing area of Brazil [33], whereas *B. phoenicis* s.s., the vector of CiLV-N, has been only found in small, private orchards located at high altitudes, where mild temperatures prevail [16]. This study reports, for the first time, the occurrence of *B. azores* in citrus in Brazil. Interestingly, individuals of this species, which are closely related to *B. phoenicis*, were only found in small orchards in the southmost subtropical region of Brazil, where temperatures are often milder. Therefore, similarly to what was observed with CiLV-N, it is predicted that CiBSV will remain restricted to a specific geographic region, mainly due to the limited distribution of its vector. On the other hand, *B. yothersi* is widely distributed from Mexico to Argentina, and that explains, at least in part, why CiLV-C, the virus it transmits, is spread throughout Latin America [34,35]. *B. yothersi* is also associated with the transmission of the dichorhavirus CiCSV, which is, so far, found only in the gardens of Teresina, Piauí, and far from the citrus-producing areas of Brazil [14,15]. The reason for the predominance of populations of *B. yothersi* in commercial citrus orchards is not known; however, the differential distribution of *Brevipalpus* species in Brazil may be associated with (i) climatic conditions, (ii) successful colonization of different citrus species and varieties, and/or (iii) different levels of susceptibility to chemical products [17,31,36,37]. Overall, the behavior of *Brevipalpus* genetic diversity in the face of climatic factors, as well as its resistance/susceptibility to acaricides, is largely unknown [36,38,39], but data suggest that *B. yothersi* may be more plastic than other species in adapting to many biotic and abiotic conditions.

In this work, we identified new viruses causing CL in sweet orange trees in southern Brazil. According to the symptoms observed in the infected plants, the morphology of the virions, genomic organization, and the nucleotide sequence identity, the three isolates were considered members of a tentative new species of the genus *Dichorhavirus*. The study also describes, for the first time, the vector activity of *B. azores* and the occurrence of a dichorhavirus causing citrus leprosis in southern Brazil. Although dichorhaviruses are not currently economically important for the citrus-producing regions in the country, this report represents the third one, describing the natural infection of sweet orange plants by bi-segmented rhabdoviruses in Brazil. The study discloses the high diversity of such viruses in South America and alerts producers and authorities about the eventual risks of crop invasion by these viruses and their mite vectors.

## 4. Materials and Methods

### 4.1. Symptomatic Citrus Trees Samples

Sweet orange leaves exhibiting localized bright chlorotic lesions and, seldomly, necrotic spots, as well as fruits with necrotic circular areas of 3–5 mm in diameter, were identified in three municipalities of the southern region of Brazil, i.e., non-commercial orchards in Marquês de Souza-Mso (samples Mso01 and Mso02) (−29.207780° and −52.235307°) and an experimental area of Embrapa Trigo, in Passo Fundo-PFd (samples PFd01 and PFd02) (−28.227545° and −52.403416°), in the state of Rio Grande do Sul (RS), and small commercial orchards in Seara-Ser (sample Ser01) in the state of Santa Catarina (SC) (−27.109039° and −52.365192°) (Figure 1).

### 4.2. Morphological and Genetic Identification of Brevipalpus Mites

There were 35 *Brevipalpus* mites found in samples MSo01 and MSo02, PFd01 and PFd02, and Ser01 that were analyzed by scanning electron microscopy (SEM) or mounted for differential interference contrast microscopy (DIC) in the laboratory “Professor Elliot Watanabe Kitajima” at ESALQ/USP, Piracicaba, SP, Brazil. The morphological characteristics of the mites and their taxonomic identification were assessed following previously described criteria [14,25]. For genetic identification, total DNA extracts from four mites found in the symptomatic samples PFd were individually extracted using a Dneasy Blood and Tissue Kit (Qiagen, HI, DE). Fragments from the mitochondrial cytochrome oxidase subunit I (COI) gene were amplified by PCR [40] and sequenced by the Sanger method. Nucleotide sequences were compared with the reference ones using Geneious Prime software version 2002.2 [41].

### 4.3. Virus Transmission Mediated by Brevipalpus Mites

*Brevipalpus* mites of different life cycle stages, collected from sample PFd02, were transferred to 10 healthy *Arabidopsis thaliana* L. plants (10 mites per plant) and maintained for 10 days, as previously described [42]. The assay was performed 3 times, at different periods during 2021, totaling 100 mites. Additionally, groups of 45 deutonymphs from isoline populations of *B. papayensis*, *B. yothersi*, *B. californicus* s.l., and *B. obovatus*, originally collected on plants of *Coffea arabica* L., *Citrus sinensis*, *Dendrobium* sp. and *Solanum violifolium*, respectively, and reared in the laboratory, were kept on symptomatic PFd02 orange leaves for 36 h for acquisition. Mites of each species that fed on the source of inoculum were transferred in groups of five per tested plant to four Arabidopsis plants, two sweet orange trees, and two common bean (*Phaseolus vulgaris*) plants, as previously described [14,43]. After symptom observation, from 5 to 10 days after infestation, the presence of the virus was detected by RT-PCR using a set of primers described in this study (Table 1). In plants where no symptoms were observed, discs with 100 mg of each inoculated leaf were used for RT-PCR assays. RT-PCR amplicons were purified by Wizard^®^ SV Gel and PCR Clean-Up System ((Promega, Madison, WI, USA), ligated into the pGEM-T-easy vector (Promega, Madison, WI, USA), and transformed in *Escherichia coli* DH10β. Plasmids from two clones of each sample were Sanger sequenced and analyzed using Geneious Prime software [41].

### 4.4. TEM and Molecular Analyses

Ultrathin sections of symptomatic leaves of sweet orange samples (MSo01 and PFd01) and the Arabidopsis plants, used as experimental hosts in transmission assays, were analyzed by transmission electron microscopy (TEM), as previously described [14]. Around 100 mg of symptomatic tissue was macerated in the presence of liquid nitrogen and used for total RNA extraction using TRIzol™ Reagent (Thermo Fisher Scientific, Madison, WI, USA). cDNAs were generated with RevertAid H Minus First-Strand cDNA Synthesis Kit (Thermo Scientific, Madison, WI, USA). Attempts to detect each known CL-associated virus and other *Brevipalpus*-transmitted viruses were performed by PCR using 3 µL of the cDNA solutions, GoTaq G2 Master Mix Green kit (Promega, Madison, WI, USA), and specific primer pairs (Table 1). The amplicons were observed on 1% agarose gels stained with ethidium bromide (10 mg/mL).

### 4.5. High Throughput Sequencing, Validation, and Annotation of Viral Genomes

RNA extracts from samples MSo01 and Ser01 were used to set up libraries for RNA by high throughput sequencing. Ribosomal RNA depletion, library construction, and sequencing were performed in the Laboratory of Animal Biotechnology (ESALQ/USP, Piracicaba, SP, Brazil) using HiSeq 2500 technology (2 × 150 nts paired-end reads) (Illumina, San Diego, CA, USA). Reads of 80–90 nts were assembled with SPAdes (k-mer 33-43-55) [49]. Viral contigs were identified using the Basic Local Alignment Search Tool (BLASTx and BLASTn) and customized plant viral genome databases retrieved from NCBI Virus (https://www.ncbi.nlm.nih.gov/labs/virus/vssi/) (accessed on 12 May 2021). Contigs were selected based on their identity with other dichorhaviruses, and the largest ones corresponding to dichorhavirus RNA1 and RNA2 were used as references for reads mapping using Bowtie2 [50]. After the assemblage of the viral genome, 25 primer pairs were designed along the genome with Geneious software [41], aiming to validate the sequences and obtain their extremities by using SMARTerR RACE 5′/3′ Kit (Clontech Laboratories, Mountain View, CA, USA) (Appendix A). Besides the cDNAs from samples MSo01 and Ser01, cDNA from the sample PFd01 was also used for genome amplification and RACE analyses. For genome validation assays, PCR amplification thermal cycles were as follows: 94 °C, 3 min; (94 °C, 30 s; 54 °C, 30 s; 72 °C, 80 s) × 35 cycles and 72 °C 5 min. Amplicons were purified by Wizard^®^ SV Gel and PCR Clean-Up System and sequenced by the Sanger method. In RACE assays, amplicons were gel-purified, ligated into the pGEM-T-easy vector, and transformed in *E. coli* DH10β. Plasmids from five of the recombinant colonies were used for Sanger sequencing. All sequences were analyzed in the Geneious Prime software [41]. Similarity plots among genome sequences obtained in this study, as well as those of dichorhaviruses available in the NCBI, were generated using SimPlot version 3.5.1 [51].

Viral open reading frames (ORFs) were identified using ORF Finder software (https://www.ncbi.nlm.nih.gov/orffinder/) (accessed on 13 May 2021). Deduced proteins, their conserved domains, signal peptides, and predicted subcellular localization were analyzed using the MOTIF Search (https://www.genome.jp/tools/motif/) (accessed on 13 May 2021), TMHMM Server version 2.0 [52], SignalP-5.0 [53], and DeepLoc version 17 [54] programs, respectively.

### 4.6. Detection of Recombination and Reassortment Events

Putative recombination events in the viral genomes obtained in this study (n = 3 genomes) and others available at the GenBank (n = 17 genomes) were assessed using seven methods (RDP, GENECONV, Bootscan, MaxChi, Chimaera, SiScan, and Topal) implemented in the RDP version 5.5 [55] and GARD [56]. Sequences were aligned using the MUSCLE program, implemented in Geneious software [41]. Recombination events were considered when detected by more than three programs (*p* ≥ 0.05). The presence of reassortment events was estimated by the topological comparison of the phylogenetic trees generated with sequences present in dichorhavirus RNA1 and RNA2 molecules. 

### 4.7. Phylogenetic Analysis Based on N and L Proteins of Plant Rhabdoviruses

All dichorhavirus sequences available at GenBank were retrieved for the alignments. Alignments were performed using the MUSCLE program that was implemented in MEGA version 7.0.21 [57]. The best models for nucleotide substitutions, determined according to likelihood ratio tests in MEGA version 7.0.21 [57], were as follows: GRT + G+I for RNA1 and RNA2 and JTT + G + I and LG + G + F for the amino acid sequences alignments of N and L proteins, respectively. Phylogenetic trees were estimated by Bayesian inference using a variant of the Markov Chains Monte MCM method (MCMC), with 6 million generations and with MrBayes implemented in the Geneious Prime software [41]. The sequences of the betanucleorhabdovirus Sonchus yellow net virus (SYNV, *Betanucleorhabdovirus retesonchi*, GB accession number: NC001615.3) and the varicosavirus lettuce big-vein associated virus (LBVaV, *Varicosavirus lactucae*, GB accession number: JN710440.1) were used as outgroups in the generated trees. The trees were viewed and edited in iTOL [58].

### 4.8. Specific Detection of New Dichochaviruses by RT-PCR

One of the most divergent regions of each genomic segment of CiBSV, when compared with known dichorhaviruses, was selected using SimPlot software version 3.5.1 [51] as the target for two specific primers pairs for detection of the *N* and *L* genes (Table 1). Sequences were aligned using MUSCLE, which was implemented in MEGA version 7.0.21 [57], and primers were designed using Primer3web version 4.1.0 [59]. To validate the method, the primers were tested with cDNA of: (i) source plants of CiBSV identified in this work, (ii) plants used in the experimental transmission of CiBSV, (iii) different plant species, hosts of the five accepted dichorhaviruses [glory-bower infected with clerodendrum chlorotic spot virus (ClCSV), coffee infected with coffee ringspot virus (CoRSV), citrus infected with CiLV-N, CiCSV, and OFV], and (iv) corresponding healthy controls of each plant species mentioned in iii. PCR thermal cycles were as follows: 94 °C, 3 min; 35 cycles of 94 °C, 30 s; 56 °C, 30 s; 72 °C, 30 s; a final extension at 72 °C for 5 min. To confirm the specificity of primers, the amplicons from samples PFd02 and MSo02 were resolved on a 1% agarose gel, purified by Wizard^®^ SV Gel and PCR Clean-Up System, and sequenced by the Sanger method.

## Figures and Tables

**Figure 1 plants-12-01371-f001:**
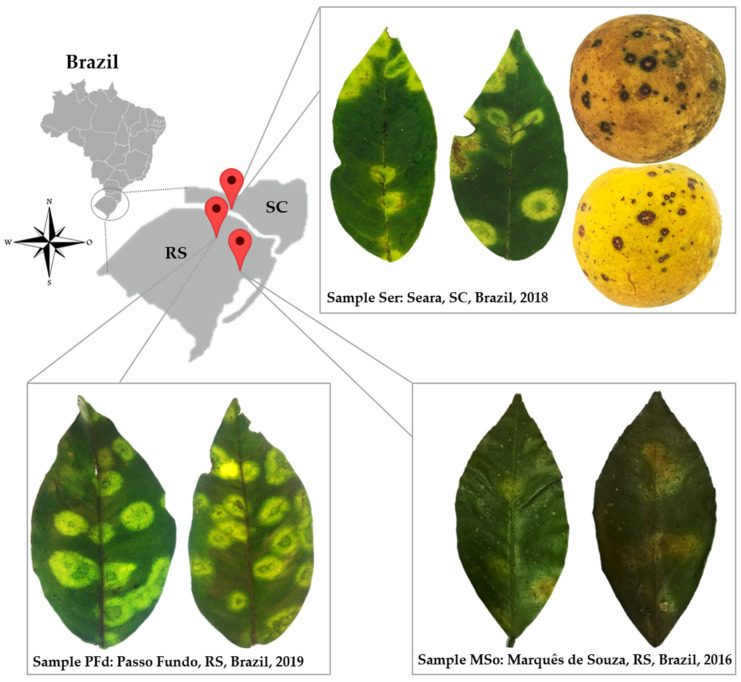
Chlorotic and necrotic symptoms in leaves and fruits of sweet orange trees (*Citrus sinensis*) collected in three localities of the southern region of Brazil. SC: State of *Santa Catarina*; RS: State of *Rio Grande do Sul*.

**Figure 2 plants-12-01371-f002:**
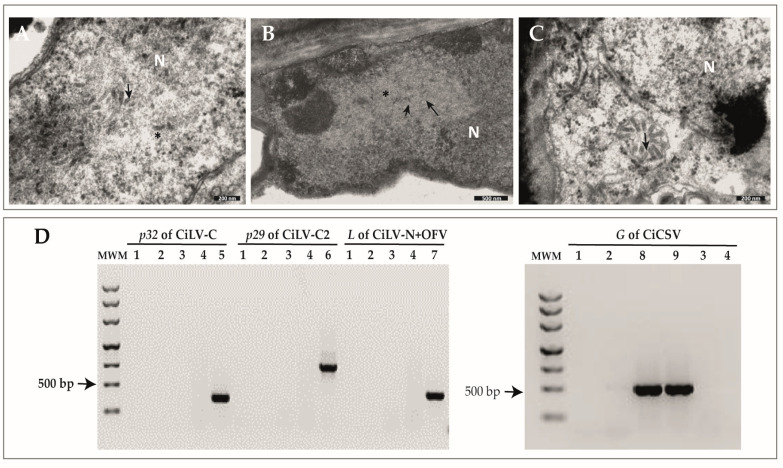
Transmission electron microscopy (TEM) and RT-PCR results from sweet orange (*Citrus sinensis*) tissues, exhibiting symptoms of citrus leprosis disease, collected in small citrus orchards in the Brazilian southern region. (**A**) Micrograph of a section of a leaf collected in Seara, Santa Catarina (SC) showing a group of particles arranged as spoke wheel inclusions (arrow) in the nucleus (N) of a palisade parenchyma cell, exhibiting an electron-lucent viroplasm (*). (**B**) Micrograph of a section of a leaf collected in Passo Fundo, Rio Grande do Sul (RS) showing the nucleus (N) with electron-lucent viroplasm (*), with some dispersed particles under formation (arrow) in parenchyma cells. (**C**) Spoke wheel inclusion (arrow) associated with the endoplasmic reticulum near the nucleus (N) in a section of a leaf collected in Marquês de Souza, RS. (**D**) The 1% agarose gel electrophoresis of RT-PCR products for the detection of dichorhaviruses. MWM: Molecular weight marker, M1181 Ladder (Sinapse Biotechnology, São Paulo, SP, Brazil). Lane 1: reverse-transcription blank; 2–4: chlorotic lesion from sweet orange leaves collected in: Seara (lane 2); Marquês de Souza (lane 3), Passo Fundo (lane 4); lanes 5–8: control plants infected by CiLV-C (expected amplicon size: 322 bp, lane 5); CiLV-C2 from Colombia (expected amplicon size: 795 bp, lane 6); CiLV-N (expected amplicon size: 362 bp, lane 7); CiCSV (expected amplicon size: 500 bp, lanes 8 and 9).

**Figure 3 plants-12-01371-f003:**
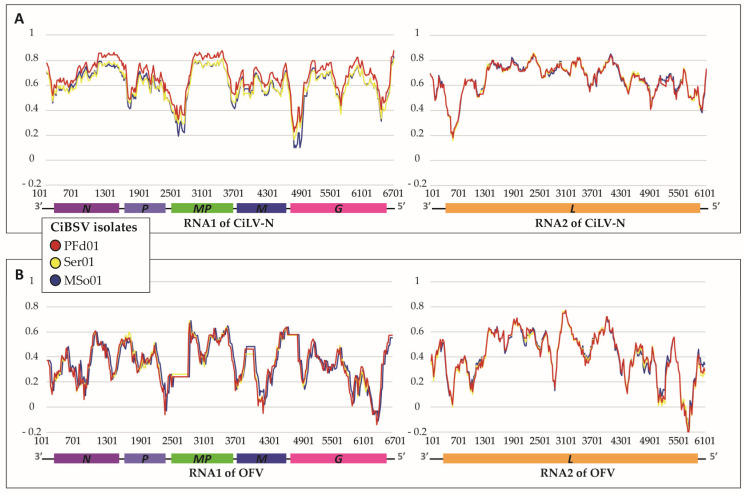
Nucleotide sequence identity of RNA1 and RNA2 of CiBSV and the genomic sequences of CiLV-N (**A**) and OFV (**B**). Data were generated by Simplot version 3.5.1. Window size: 200 nts. Step size: 20 nts. The vertical axis indicates the nucleotide identities, expressed as percentages, and the horizontal axis indicates the nucleotide positions of RNA1 and RNA2 of CiLV-N (**A**) or OFV (**B**).

**Figure 4 plants-12-01371-f004:**
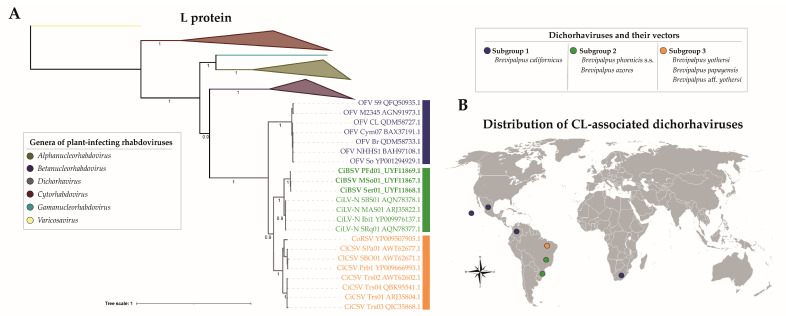
Phylogenetic reconstruction of CiBSV isolates (bold) and members of the genus *Dichorhavirus*, based on L protein sequences (**A**) and distribution of citrus leprosis (CL)-associated dichorhaviruses and their *Brevipalpus* vectors (**B**). The trees was generated by Bayesian inference using MrBayes with 2,000,000 generations, and the varicosavirus lettuce big-vein associated virus (LBVaV) was used as an outgroup. Figures near the main nodes indicate the posterior probability values. The color of vertical bars, at the right, depicts each phylogenetic subgroup of dichorhaviruses and indicates the identified mite vectors, as described in the table. The phylogenetic tree was edited using iTOL. B. The distribution of CL-associated dichorhaviruses was based on literature data [3,11,14] and the results obtained in this study.

**Figure 5 plants-12-01371-f005:**
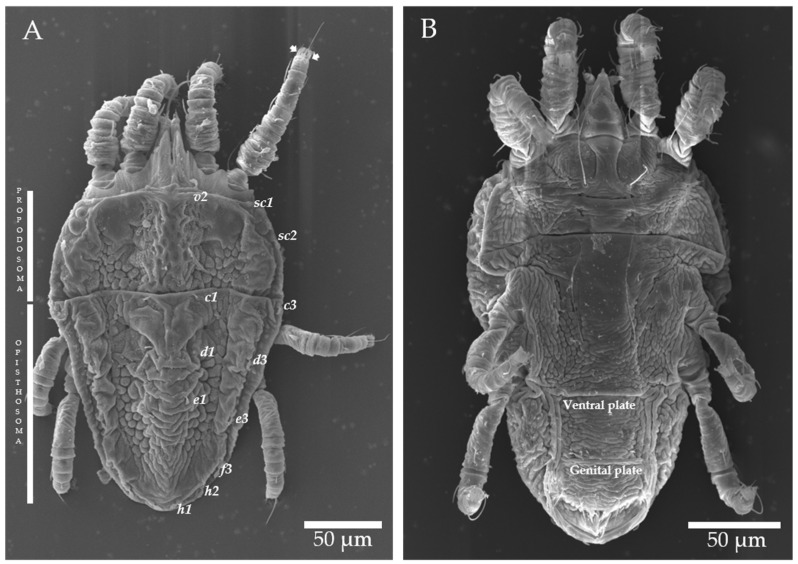
Microphotographs of *Brevipalpus azores* mites found in leaves and fruits of sweet orange (*Citrus sinensis*) collected in Marquês de Souza, RS (**A**) and Seara, SC (**B**). Evaluated morphological characteristics were as follows: number of dorsolateral setae, number of solenidia on the tarsus of leg II (setae), and reticulation patterns of propodosoma and opisthosoma. Ventral and genital shield reticulations can be observed in the center.

**Figure 6 plants-12-01371-f006:**
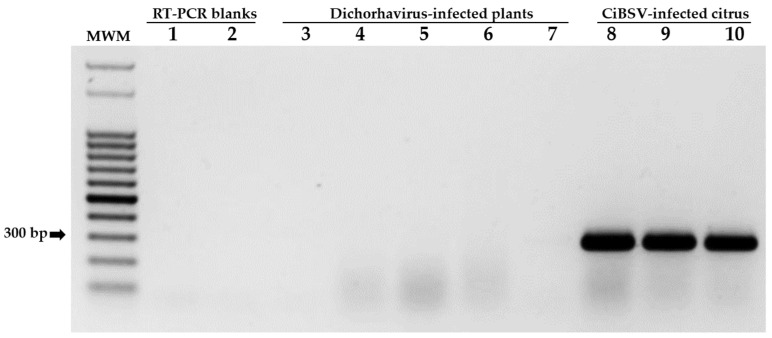
The 1% agarose gel electrophoresis of RT-PCR products using specific pair primers for detection of the *N* gene of CiBSV. MWM: Molecular weight marker, 100 bp DNA Ladder (Madison, WI, USA); lane 1: reverse-transcription blank; lane 2: healthy citrus plant; lane 3: ClCSV-infected glory-bower plant from Brazil; lane 4: CoRSV-infected coffee from Brazil; lane 5: CiLV-N-infected citrus from Brazil; lane 6: CiCSV-infected citrus from Brazil; lane 7: OFV-infected citrus from Mexico; lane 8: citrus plant infected with CiBSV isolate PFd02; lane 9: citrus plant infected with CiBSV isolate MSo02; lane 10: citrus plant infected with CiBSV isolate Ser01.

**Table 1 plants-12-01371-t001:** Primers used for the detection of *Brevipalpus* mites-transmitted viruses by RT-PCR.

Genus	Virus ^1^	Target	Sequence 5′-3′		Amplicon Size (bp)	Reference
** *Dichorhavirus* **	OFV	*N*	F:	TGTCATAGCCGACATAAACACC	326	[44]
			R:	TGTAGAGCTTGCGAGATACAGG		
	CiLV-N	*N*	F:	CCGTACCCATTGTGAAAATA	420	[16]
			R:	GAACCCCTTTGAGGAATG		
	OFV + CiLV-N	*L*	F:	CAASTGTCATGCCTGCATGG	362	
			R:	TTGATRCATGATGCRAGRCTGTATG		
	CiCSV	*G*	F:	CTGTTTTGCCCATGCTAC	500	[14]
			R:	CCTCCTCTTCTAGCGTCAT		
	CoRSV	*L*	F:	GGACCATGAGACAGGAGGTG	394	[45]
			R:	CTCTGCCAGTCCTCAATGTG		
	ClCSV	*L*	F:	AGTGTACCGCCTCACAGAAG	219	[46]
			R:	CGGGGTCTTGTTGTTCATAG		
**Tentative dichorhavirus**	CiBSV	*N*	F:	CAGTCACTATAGATTACTCAGCAG	296	This study
			R:	TGCTACTCCTACCATCAC		
		*L*	F:	GCCTACGGAGAGGAAGAT	938	
			R:	GGGTCTACTGAGCTGTATATGA		
** *Cilevirus* **	CiLV-C	*p24*	F:	CGCAGTTTCCTAATAACACC	322	[20]
			R:	GGGAGTTCAGCATAAAGC		
	CiLV-C2	*p29*	F:	ATGAGTAACATTGTGTCGTTTTCGTTGT	795	[47]
			R:	TCACTCTTCCTGTTCATCAACCTGTT		
	PfGSV	*p29*	F:	ACACCAAGAGTACTATCGATC	452	[48]
			R:	CATCAAGTGGAGCAAGTTC		

^1^ Dichorhaviruses: orchid fleck virus (OFV); citrus leprosis virus N (CiLV-N); citrus chlorotic spot virus (CiCSV); coffee ringspot spot virus (CoRSV); clerodendrum chlorotic spot virus (ClCSV); the tentative dichorhavirus identified in this study: citrus bright spot virus (CiBSV); cileviruses: citrus leprosis virus C (CiLV-C), citrus leprosis virus C2 (CiLV-C2), and passion fruit green spot virus (PfGSV).

## Data Availability

The virus sequences derived from this study can be found in GenBank under the accession numbers: MZ773933 and MZ773938 (CiBSV_PFd01); MZ773934 and MZ773936 (CiBSV_MSo01); MZ773935 and MZ773937 (CiBSV_Ser01).

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
