# Peer review of "Citrus Bright Spot Virus: A New Dichorhavirus, Transmitted by Brevipalpus azores, Causing Citrus Leprosis Disease in Brazil"

_plants, 2023, doi:10.3390/plants12061371_

Round 1
Reviewer 1 Report
The authors described a new virus species associated with citrus leprosis in sweet orange in Brazil. The infected plants displayed a variation of characteristic symptoms, and the authors showed that, although the general genomic organization was typical of other dichorhaviruses known to cause citrus leprosis, the nucleotide and amino acid analyses confirmed that it was a new species. Phylogenetic analyses conducted on the genomic sequences additionally showed the isolates of the new species grouping separately from known species, confirming the new species status. Finally, the authors showed through transmission assays that only the Brevipalpus azores mite is able to transmit the virus to Arabidopsis plants.
The manuscript is well-written and the conclusions made are supported by the results presented here. There are a few minor typos/edits that I think need to be addressed, and I have highlighted these throughout the text.
However, there is one area that is not clear to me regarding the transmission studies using B. papayensis, B. obovatus, B. californicus s.l., and B. yothersi mites. Lines 356 -359: "Mites of each species that fed onto the source of inoculum were transferred in groups of five per tested plant to four Arabidopsis plants, two sweet orange trees, and two common beans (Phaseolus vulgaris) plants, as previously described [14,43]". I have not seen the results for this part of the assay mentioned in text. Instead, virus transmission (or failure to) of the other Brevipalpus species is only mentioned for Arabidopsis, but not for sweet orange and common beans.

Author Response
The requested results were added. New data were included, e.g.: partial sequences of CiBSV from infected Arabidopsis and sweet orange and their GenBank accession numbers.

Reviewer 2 Report
Line 19, 46-47, 76, 271, 350, 354 - write the authors when 1st mentioned
Line 60-61 – „as is suggested by phylogenetic analyses 60 suggest“ – say that in a better way
Line 94 – put space between the words: „the parenchymal“
Line 355 – Laboratory – with uppercase
Table 1 – column 1 – better arrange the data not to overlap one over the other
Author Response

(The authors gave the same response as above.)
